METHODS

# Learning genetic perturbation effects with variational causal inference

Emily Liu[1,2☯], Jiaqi Zhang[1,2,3☯*], Caroline Uhler[1,2,3*]

**1** Department of Electrical Engineering and Computer Science, MIT, Cambridge, Massachusetts, United States of America, **2** Eric and Wendy Schmidt Center, Broad Institute, Cambridge, Massachusetts, United States of America, **3** Laboratory for Information and Decision Systems, MIT, Cambridge, Massachusetts, United States of America

☯ These authors contributed equally to this work.
* viczhang@mit.edu (JZ); cuhler@mit.edu (CU)

## Abstract

Advances in sequencing technologies have enhanced the understanding of gene regulation in cells. In particular, Perturb-seq has enabled high-resolution profiling of the transcriptomic response to genetic perturbations at the single-cell level. This understanding has implications in functional genomics and potentially for identifying therapeutic targets. Various computational models have been developed to predict perturbational effects. While deep learning models excel at interpolating observed perturbational data, they tend to overfit in the lack of enough data and may not generalize well to unseen perturbations. In contrast, mechanistic models, such as linear causal models based on gene regulatory networks, hold greater potential for extrapolation, as they encapsulate regulatory information that can predict responses to unseen perturbations. However, their application has been limited to small studies due to overly simplistic assumptions, making them less effective in handling noisy, large-scale single-cell data. We propose a hybrid approach that combines a mechanistic causal model with variational deep learning, termed Single Cell Causal Variational Autoencoder (SCCVAE). The mechanistic model employs a learned regulatory network to represent perturbational changes as shift interventions that propagate through the learned network. SCCVAE integrates this mechanistic causal model into a variational autoencoder, generating rich, comprehensive transcriptomic responses. Our results indicate that SCCVAE exhibits superior performance over current state-of-the-art baselines for extrapolating to predict unseen perturbational responses. Additionally, for the observed perturbations, the latent space learned by SCCVAE allows for the identification of functional perturbation modules and simulation of single-gene knockdown experiments of varying penetrance, presenting a robust tool for interpreting and interpolating perturbational responses at the single-cell level.

**Data availability statement:** Code and relevant data for the SCCVAE model and experiments are available at https://github.com/uhlerlab/sccvae.

**Funding:** This work was carried out with major support from the Eric and Wendy Schmidt Center at the Broad Institute of MIT and Harvard. E.L. was supported by the Eric and Wendy Schmidt Center at the Broad Institute. J.Z. was partially supported by an Apple AI/ML PhD Fellowship. C.U. was partially supported by NCCIH/NIH (1DP2AT012345), ONR (N00014-22-1-2116 and N00014-24-1-2687), the United States Department of Energy (DE-SC0023187), the Eric and Wendy Schmidt Center at the Broad Institute, and a Simons Investigator Award. The funders had no role in study design, data collection and analysis, decision to publish, or preparation of the manuscript.

**Competing interests:** The authors have declared that no competing interests exist.

## Author summary

Understanding how genes interact and respond to perturbations is crucial for uncovering the mechanisms of cells and identifying potential ways to treat diseases. Recent advances in sequencing technologies now allow us to measure how individual cells react when specific genes are altered. However, making sense of this complex data requires advanced computational tools. In our work, we address the challenge of predicting how cells respond to potentially new untested genetic perturbations. We noticed that while deep learning models perform well on data measured before, they struggle with making predictions on new cases. On the other hand, models based on biological understanding can, in theory, make better predictions, but they often rely on overly simple assumptions that do not hold with real-world data. We developed a new method that combines the strengths of both approaches. Our model, called SCCVAE, uses knowledge of gene networks together with deep learning to better predict how cells will respond to gene changes. It can simulate new experiments and help identify groups of genes that work together. This tool could be valuable for researchers studying perturbational changes, as well as gene functions and diseases.

## 1 Introduction

Gene-editing technologies provide useful probes for the study of gene regulation in cells [1]. By perturbing individual genes and observing transcriptomic changes, we can disentangle and resolve the downstream effects of these perturbations. These insights facilitate a range of downstream applications, from identifying genes involved in fundamental cellular processes (e.g., [2–4]) to discovering potential drug targets for therapeutic use (e.g., [5,6]). There are numerous potential target genes for perturbation. Perturb-seq allows for large-scale exploration by combining high-throughput CRISPR gene editing with single-cell RNA sequencing [7,8]. Recent advances have further expanded its scale, enabling the collection of data on genome-wide perturbations in millions of cells [9]. Understanding cellular responses to the genetic perturbations introduced through these high-content data is of great importance.

Various computational approaches have been proposed to interpret and predict perturbational effects. One predominant line of work explores the performative powers and inductive biases brought by popular deep learning architectures. For example, [10] uses a compositional architecture combined with an adversarial network to disentangle perturbational effects from the basal cell states. [11] utilizes two separate autoencoders to learn perturbation-specific and cell-specific latent representations and employs a normalizing flow to map between these representations. [12] uses a graph-based network that leverages a gene ontology-based graph to simultaneously learn perturbation embeddings and their corresponding effects. [13] proposes a modified variational autoencoder with carefully designed noise models, modeling perturbational effects as sparse shifts of these noise distributions. [14] uses a variational autoencoder with a graph attention architecture to encode gene regulations. [15] proposes to model single-cell transcriptomics data using a diffusion model

with cross-attentions that incorporate disentangled concept embeddings. The concept embeddings are constrained by a predefined causal graph that encodes the causal relationships between concepts (e.g., tissues, cell types, perturbations), which enables counterfactual generation upon changing these concepts. Note that this causal graph differs from that of gene regulation described above, as it models the relationships between hierarchical concepts. Several variational-inference-based analyses have also been proposed to interpret the observed perturbations [16,17]; see [18] for a comprehensive review. Recently, transformer architectures have also been used to learn unsupervised representations of cells and genes, with perturbation prediction being a downstream task [19,20]. While these methods excel at interpolating observed data, they are prone to overfitting, which may limit their ability to generalize to unseen perturbations. [21] observed that simple linear models can outperform sophisticated deep learning models in generalizing to unseen perturbations. However, their approach focuses on pseudo-bulk analysis, whereas this work aims to resolve perturbational effects at the single-cell level.

There are several axes to consider when generalizing to unseen perturbations. *First*, the generalization task may involve extrapolating from single-gene perturbations to combinatorial perturbations that target multiple genes or to single-gene perturbations targeting novel genes, potentially administered at varying penetrance / multiplicity of infection. In this work, we mainly focus on extrapolations to novel single-gene perturbations. *Second*, depending on the task, the principles one uses for generalization may differ. For example, to generalize to combinatorial perturbations, a popular principle is to use additivity to combine the effects (potentially in a latent space) and learn the interactions as residuals (e.g., [12,22–24]). To generalize to unseen novel perturbations, two major principles are to use prior knowledge of how the new target relates to observed targets (e.g., [11,12] reviewed above) or a mechanistic model that specifies how the perturbation propagates according to a gene regulatory network (e.g., [25]). Prominent deep learning models, as discussed above, are mostly prior-knowledge-based, typically relying on gene ontology annotations, which might suffer from poor generalization as such annotations usually capture partial correlations. On the other hand, mechanistic models hold greater potential for extrapolation as they capture regulatory information. Prior mechanistic models usually make assumptions about how perturbations change transcriptomic profiles. For example, [25] used a linear model based on a pre-fixed gene regulatory network inferred from ChIP-seq data and demonstrated its utility for recommending transcription factors for direct differentiation. [26] employed a linear causal model based on a learned gene regulatory network obtained via causal discovery algorithms and showed its usefulness for intervention design in a semi-synthetic experiment. [27] uses a chemical master equation to simulate transcription. However, their applications have been limited to relatively small studies due to simplistic parametric assumptions and/or expensive stochastic differentiation equation simulations.

In our work, we propose a hybrid model that combines a mechanistic causal model with variational deep learning, termed the Single Cell Causal Variational Autoencoder (SCCVAE). The mechanistic causal model captures regulatory information by employing a learned regulatory network and models perturbations as shift interventions that propagate through this network. It is defined on a lower-dimensional space, capturing essential information to reconstruct the entire transcriptomic readout. To address the issue of simplistic parametric assumptions in most mechanistic models, SCCVAE integrates this model into a variational autoencoder. This integration allows SCCVAE to learn and generate rich, comprehensive transcriptomic responses. Our findings demonstrate that SCCVAE excels at generalizing beyond observed perturbations, enabling accurate predictions of unseen single-gene perturbations, and outperforms both standard and state-of-the-art baselines. The mechanistic model specifies the penetrance of the perturbation, allowing simulation of single-gene knockdown perturbations with varying penetrance. As for the observed perturbations, since SCCVAE learns how the perturbation shifts the variables defined by the mechanistic model, we can extract this information to serve as a perturbation representation, which we observe to capture functional perturbation modules.

## 2 Methods

### 2.1 Variational causal model for perturbations

Consider the gene expressions of a cell, denoted by $X^p$, perturbed by a single-gene perturbation (i.e., single-guide RNA) represented by $p$. When the cell is a negative control (e.g., with non-targeting guide RNAs), we denote its gene expressions as $X^\varnothing$ or $X$ in short with $p = \varnothing$. Each cell is measured by the expression levels of $m$ distinct genes, i.e., $X^p \in \mathbb{R}^{m \times 1}$. We now explain the components of SCCVAE.

**2.1.1 Structural causal model.** Central to the design of our model is the concept of structural causal models [28,29]. A *structural causal model* (SCM) is a framework for understanding complex systems by modeling the regulations between causal variables. Formally, a structural causal model is defined as a tuple $\langle Z, U, F \rangle$ consisting of a set of exogenous noise variables $Z$, a set of endogenous random variables $U$, and a set of functions $F : Z \times U \to U$. Each SCM is associated with a directed acyclic graph $\mathcal{G}$, where node $i$ in $\mathcal{G}$ corresponds to an endogenous variable $U_i$ and edges denote causal relationships between variables. For a given $i$, the endogenous variable $U_i$ depends on its associated exogenous noise variable $Z_i$ and its endogenous parents $U_{\mathrm{pa}(i)}$, where $j \in \mathrm{pa}(i)$ if there is an edge $j \to i \in \mathcal{G}$. Mathematically, we have

$$U_i = F_i(U_{\mathrm{pa}(i)}, Z_i). \tag{1}$$

In the context of gene regulations, the endogenous variables $U \in \mathbb{R}^{n \times 1}$ denote an abstracted representation of $n$ gene modules, where each $U_i$ corresponds to one gene module. Intuitively, this representation can be interpreted as a low-dimensional feature summarizing the expression of $m$ genes in the context of a cell, where $m$ can be much larger than $n$. The causal graph $\mathcal{G}$ indicates gene regulations, where $j \in \mathrm{pa}(i)$ if genes in module $j$ directly regulate genes in module $i$. The exogenous variables $Z$ record the intrinsic variations of $U$, not explained by regulations. Given the values of $U_{\mathrm{pa}(i)}$ and $Z_i$, function $F_i$ then generates $U_i$ associated with module $i$. In our setting, we choose to use a specific parametric family of SCM, where $Z_i$ are assumed to be Gaussian and $F_i$ are linear functions, as this parameterization can encapsulate all multivariate Gaussian distributions, which we observe to sufficiently fit the transcriptomic data. Here the Gaussian family is used for the latent space, which is linked to the observed transcriptomic space through a decoder that can flexibly transform a latent Gaussian distribution into the appropriate family of distributions in the output space. Prior work have also used Gaussian-based latent modeling of single-cell data [30,31]. In other words, Eq (1) is realized by

$$U_i = \sum_{j \in \mathrm{pa}(i)} A_{ij} U_j + Z_i, \quad Z_i \sim \mathcal{N}(0, \sigma_i^2), \tag{2}$$

where $A_{ij}$ describes the contribution of $U_j$ in $U_i$. Since the scaling of the endogenous variables can be set arbitrarily, we further assume that $\sigma_i = 1$.

A genetic perturbation $p$ alters the endogenous variables $U$ into $U^p$ by modifying the regulatory relationships in Eq (2). Here, we use an additive shift model, summarized by

$$U_i^p = \sum_{j \in \mathrm{pa}(i)} A_{ij} U_j^p + Z_i + S_i^p, \quad Z_i \sim \mathcal{N}(0, \sigma_i^2), \tag{3}$$

where $S^p$ is the shift vector associated with perturbation $p$. Fig 1A shows an example. In the following section, we explain how to learn this SCM within a variational inference framework.

We make a note here that the definition of SCM assumes acyclic $\mathcal{G}$ to ensure that Eq (1) has a unique solution. However, it can be easily extended to cyclic graphs to model feedback loops by setting $\mathrm{pa}(i)$ as all parent nodes of $i$ excluding itself in the cyclic graph. When the causal variables are observed, methods for causal discovery in acyclic models

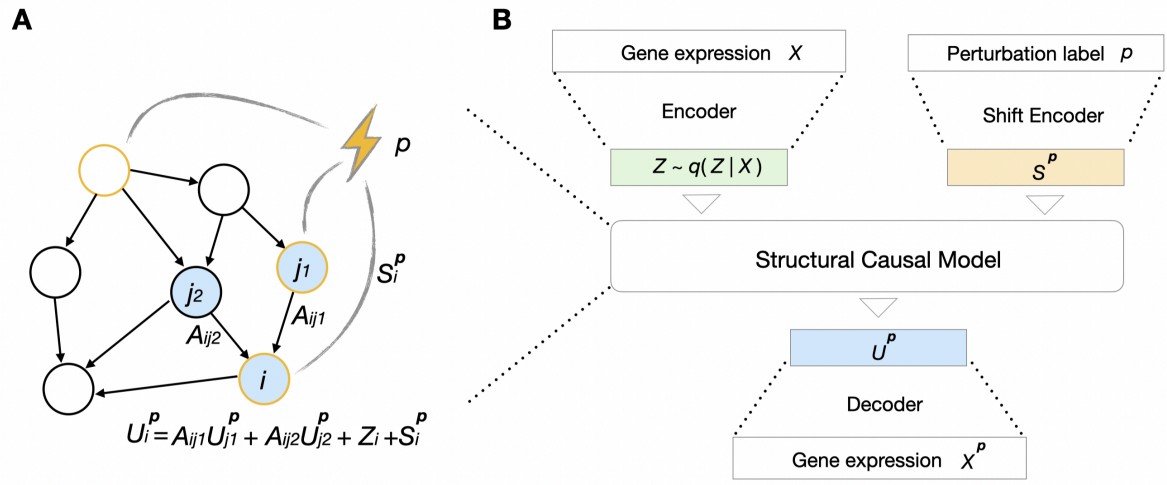

**Fig 1**. **Key components of SCCVAE.** (A) Illustration of a structural causal model, where the parameters associated with gene $i$ are annotated. (B) The architecture of SCCVAE. It contains: an expression encoder that maps $X$ to exogenous noise variables $Z$, a shift encoder that maps $p$ to a shift vector $S^p$, a structural causal model (e.g., as illustrated in (A)) that maps $Z, S^p$ to $U^p$, and an expression decoder.

have been extended to cyclic settings, including constraint-based algorithms (e.g., [32–34]) which generalize conditional independence tests from d-separation to $\sigma$-separation, and score-based algorithms (e.g., [35,36]).

**2.1.2 Hybrid variational causal model.** SCCVAE integrates and learns the SCM in a variational autoencoder. It consists of four key components: the *expression encoder*, the *shift encoder*, the *SCM*, and the *expression decoder*. Fig 1B shows all inputs, outputs, and components of the model architecture. We discuss identifiability of the corresponding data-generating process in Note A in S1 Text.

The *expression encoder* takes in the expression of one control cell $X \in \mathbb{R}^{m \times 1}$ and encodes it into the exogenous variables $Z \in \mathbb{R}^{n \times 1}$, with $n < m$. Intuitively, such $Z$ represents intrinsic variations of $n$ gene modules, not explained by regulations, that is shared between the control and the perturbed distributions. This set of $n$ modules captures essential information to reconstruct the entire transcriptomic readout. (In our implementations, we select $n = 512$.) We encode the conditional distribution $q(Z \mid X)$ using diagonal normal distribution, where the mean and variance are parameterized by neural networks, and minimize the KL divergence $\mathbb{E}_X \left( D_{KL} \left( q(Z \mid X) \parallel p(Z) \right) \right)$, where $p(Z)$ corresponds to $\mathcal{N}(0, I_n)$ with $I_n$ being the identity matrix of order $n$. We then use the reparameterization trick [37] to sample $Z$ from $q(Z \mid X)$ to obtain the endogenous variables.

As inputs to the *shift encoder*, we need a label representing each single-gene perturbation. This label should be attainable for unseen perturbations for generalization purposes. Here we use the top principal components of the transposed control cell expression matrix of shape $\mathbb{R}^{m \times l}$ as perturbation labels, where $l$ is the number of control cells. For each of the $m$ measured genes, we compute the associated top principal components vector in $\mathbb{R}^l$ using the control cell expression matrix. We take top principal components in this vector as our label representing perturbation on this measured gene. This produces similar representations for genes that co-express in control cells. This choice reflects the inductive bias that genes with similar functions—and therefore likely to induce similar perturbational effects—tend to co-express under natural variations such as the heterogeneity observed in control cells. A similar prior has also been adopted in previous work [21].

The *shift encoder* encodes a perturbation label $p$ into a shift vector $S^p \in \mathbb{R}^{n \times 1}$. Intuitively, such $S^p$ denotes the direct effects that a perturbation has on these $n$ gene modules, capturing both on-target and potentially off-target effects. Such

direct effects then propagate to other endogenous variables through Eq (3). Note that Eq (2) is a special case of Eq (3) with $p = \varnothing$ and $S^p = 0$.

We can stack Eq (3) for different $i = 1, \dots, n$ and write it into a matrix form

$$U^p = AU^p + Z + S^p,$$

which is equivalent to

$$U^p = (I_n - A)^{-1}(Z + S^p),$$

where $U^p = (U_1^p, \dots, U_n^p)^\top$ and $A_{ij} = 0$ if $j \notin \text{pa}(i)$. Therefore in the *SCM*, we only need to specify the graph $\mathcal{G}$ and parametrize $A$ by masking it according to the sparsity pattern in $\mathcal{G}$, upon which $U^p$ can be directly computed based on $Z$ and $S^p$. In our implementations, we find that a learned network $\mathcal{G}$ (i.e., using an upper-triangular mask) that is optimized during training to work well.

Finally the *expression decoder* takes the perturbed endogenous variables $U^p$ and constructs the expression profile $X^p$ of perturbed cells. The model is trained using the evidence lower bound with an added maximum mean discrepancy (MMD) term [38] specified by

$$\begin{aligned}
\mathcal{L}(\theta) = \; & \mathbb{E}_p \left( \mathbb{E}_X \left( \|X^p - \hat{X}^p\|_2^2 \right) \right) + \\
& \beta \cdot \mathbb{E}_X \left( D_{KL}(q(Z \mid X) \| p(Z)) \right) + \\
& \gamma \cdot \mathbb{E}_{p \neq \varnothing} \left( MMD(X^p, \hat{X}^p) \right).
\end{aligned}$$

The evidence lower bound in the first two terms (with $p = \varnothing$) maximizes the likelihood of the hybrid causal model over the control distribution, whereas the mean square error in the first term (with $p \neq \varnothing$) and the added MMD term match and stabilize the training for the perturbational distributions.

Here $\theta$ subsumes all the unknown parameters, $\hat{X}^p$ denotes the output of SCCVAE, and hyper-parameters $\beta, \gamma$ are scaling factors specified in Note B in S1 Text.

In summation format, $\mathcal{L}(\theta) = \frac{1}{|P|} \sum_{p \in P} \frac{1}{D_p} \sum_{i=1}^{D_p} \|X_i^p - \hat{X}_i^p\|_2^2 + \frac{\beta}{D_\varnothing} \sum_{i=1}^{D_\varnothing} \left( D_{KL}(q(Z \mid X_i) \| p(Z)) \right) + \frac{\gamma}{|P|} \sum_{p \in P} MMD(X^p, \hat{X}^p)$, where $P$ is the set of all perturbations, $D_p$ is the number of cells for $p \in P$, and $D_\varnothing$ is the number of control cells.

## 2.2 Shift selection for unseen perturbations

When generalizing to an unseen single-gene perturbation, especially in knockdown or activation experiments, it is essential to account for the penetrance of the perturbation, i.e., the magnitude of its effect. This is because such perturbations may be administered with guides of different efficiencies compared to the perturbations seen during training. As a result, simply encoding the identity of the new perturbation is insufficient; we must also infer how strongly it acts. Specifically, for an unseen perturbation $q$, we can encode it into a shift vector $S^q \in \mathbb{R}^{n \times 1}$ using SCCVAE trained on observed perturbations. To quantify the effects of different penetrance, we attribute one single scalar $c^q \in \mathbb{R}$ to denote its effect on the endogenous variables. In other words,

$$U^q = AU^q + Z + c^q \cdot S^q.$$

Then $U^q$ is passed to the decoder to generate in-silico transcriptomic profiles of single cells perturbed by $q$.

If additional metadata exists, e.g., a proliferation screen with the same library, one can use this information to specify the intensity of the perturbation. When such information is not available, one can use, e.g., bulk perturbation data to select

$c^q$ in order to evaluate the capability of the model. Specifically, one can grid search for $c^q$ within a range by comparing the average predicted perturbed cell expression $\hat{X}^q$ with the bulk expression of cells subjected to perturbation $q$ using mean squared error. The accurate penetrance $c^q$ should give rise to the minimal error. This shift selection process allows for the identification of penetrance from pseudo-bulk data without having access to single-cell data for novel perturbations.

Note that such shift selections are unavoidable to obtain accurate predictions, as the unseen perturbation could have very distinct penetrance. However, the performance of the model will still rely on whether SCCVAE successfully learns the underlying regulatory information, as attributing a single scalar parameter alone will not lift sufficient capacity to generalize to unseen perturbations.

**2.2.1 Computational complexity.** Consider $N_I$ perturbations at inference time. By generating $B$ cells per perturbation, the computational complexity of evaluating $C$ possible shift values is $O(N_I B C m)$, where $m$ is the number of genes in the considered expression data. The typical choice of $B$ is around $32 \sim 128$. A finer level of shift selection with larger $C$ results in more accurate results, where the computational complexity scales linearly in $C$.

## 3 Results

We evaluate SCCVAE on the single-gene perturbational datasets by [39] which contains normalized gene expression transcripts of essential genes for two different cell lines, K562 and RPE1 cells, where $m$=8563 on K562 cells and $m$=8749 on RPE1 cells.

In our first set of analyses, we focus on the cancerous K562 cell line and examine a subset of perturbations that induced distinct expression distributions from the control distribution, in order to reduce the noise to signal ratio. This subset is chosen by first filtering out the perturbations with less than 200 single cells. Out of the perturbations with sufficient cells, we consider the subset that is distinct from controls and thus requires nontrivial predictions. This subset is identified by performing a logistic regression task to distinguish perturbed cells from negative controls. Perturbations achieving an average five-fold cross-validation score greater than 0.6 are retained, resulting in $N = 279$ perturbations. Further details can be found in Note C.1 in S1 Text.

We additionally include experiments on the entire K562 cell line (without filtering perturbations) by comparison, along with experiments on a selected subset of the RPE1 cell line. The additional experimental results can be found in Note C.7 in S1 Text.

### 3.1 Setup

**3.1.1 In-distribution experiments.** In this set of experiments, we evaluate how well different models can interpolate observed perturbations. All cells in the test set are drawn from the same perturbational distributions as those in the training set. For each perturbation in the dataset, we assign 70% of the cells to the training set, 10% to the validation set, and the remaining 20% to the test set.

**3.1.2 Out-of-distribution experiments.** In this set of experiments, we evaluate the performance of different models on generalizing to unseen perturbations. We designate the train/test split so that there is no overlap between the test set and the train set perturbations. First, we randomly select 20% of the perturbations to make up the test set. Out of the remaining perturbations, we assign 85% of the cells from each distribution to the training set, and the remaining 15% to the validation set. To account for variation in the amount of distributional shift that occurs from random selection of test perturbations, we repeat this experiment using five different train/test splits, so that each perturbation is included in the test set for one of the splits. Results are averaged across all splits. For each unseen perturbation $X^q$ (defined in Sect 2.2), we select $c^q$ by searching through the range of all learned $c^p$ in the model and select the value that minimizes mean squared error for pseudo-bulk predictions. Details of this search process are found in Note B.3 in S1 Text.

In both in-distribution and out-of-distribution tasks, training hyperparameters were selected based on minimizing loss (defined in Sect 2.1) within the validation set, drawn from the same distribution as the training set and determined as

described above. Hyperparameters are selected through the validation set separately for each train/test split. As such, no points from the test set are seen in the hyperparameter search for any given split. Further details are given in Note B in S1 Text.

## 3.2 Prediction of single-cell expressions

To evaluate SCCVAE, we consider six metrics: Mean squared error, Pearson correlation of expression change, maximum mean discrepancy, energy distance, and fraction of genes changed/unchanged from control in the opposite direction of ground truth. Formulas for computing each metric are found in Note B.4 in S1 Text. Each perturbation is evaluated separately, and we report the mean and standard deviation across all test perturbations for each metric. Each metric is computed on the set of all essential genes and on the more difficult task of predicting just the top 50 most highly variable genes.

### 3.2.1 SCCVAE versus single-cell-level baselines.
Here we focus on perturbational effects at the single-cell level. For baselines, we compare against the control cell distribution, a popular deep learning model (GEARS [12]), and a transformer-based model (scGPT [20]) for single-cell predictions.

Results of the in-distribution experiments can be found in Table A in S1 Text, where we observed that SCCVAE outperforms or is on par with both baselines on all metrics. This shows the benefit of utilizing an expressive network in SCCVAE to interpolate observed perturbations.

Table 1 shows the results of the out-of-distribution experiments. It can be seen that the SCCVAE outperforms both the GEARS and control baselines for generalizing to unseen perturbations. The improvements above baselines for single-cell based metrics, i.e., MMD and Energy Distance, are more pronounced when evaluating on all essential genes. This is further illustrated in Fig 2A–2B, where we observe that SCCVAE matches the perturbation distributions much better than the baselines. The fraction of perturbations changed or in the same direction from control is also improved in the SCCVAE predictions. SCCVAE additionally achieves better MSE and Pearson R than the baselines, although all baselines perform well over these metrics. However, in SCCVAE there is much less variation across different perturbations.

Additionally, although scGPT attains better Pearson correlations and fraction changed/same metrics, the distributional error metrics (MMD and Energy Distance) are all higher than control. This is indicative of the ability of transformer-based

**Table 1**. **SCCVAE vs baselines on the out-of-distribution task.** Results are averaged across five different splits, each containing a different set of perturbations in the test set covering the entire set of all perturbations. Both when evaluating all essential genes and top 50 genes, SCCVAE achieves better metrics than control or GEARS for every metric except Pearson correlation, where all three methods achieve similar mean values but SCCVAE has lower variance in its predictions. SCCVAE outperforms a transformer-based model, scGPT, on both distributional-based metrics. When compared to a simple linear model from Ahlmann-Eltze et al [21], SCCVAE achieves similar metrics to the linear model, while additionally achieving low distributional loss, expanding its scope beyond the linear model.

| All genes | Control | GEARS | scGPT | Linear | SCCVAE |
|---|---|---|---|---|---|
| MSE | $0.00817_{\pm 0.00575}$ | $0.00734_{\pm 0.00524}$ | $0.0101_{\pm 0.00105}$ | $\mathbf{0.00438_{\pm 0.000392}}$ | $0.00542_{\pm 0.002}$ |
| PearsonR | $-0.002_{\pm 0.054}$ | $0.160_{\pm 0.094}$ | $\mathbf{0.535_{\pm 0.0652}}$ | $0.496_{\pm 0.0413}$ | $0.498_{\pm 0.164}$ |
| MMD | $0.240_{\pm 0.0391}$ | $0.239_{\pm 0.0405}$ | $0.266_{\pm 0.0104}$ | - | $\mathbf{0.229_{\pm 0.0193}}$ |
| Energy Dist. | $0.0363_{\pm 0.0138}$ | $0.0333_{\pm 0.0132}$ | $0.0337_{\pm 0.00929}$ | - | $\mathbf{0.0299_{\pm 0.00958}}$ |
| Frac. Same | $0.496_{\pm 0.0175}$ | $0.551_{\pm 0.0590}$ | $\mathbf{0.717_{\pm 0.0468}}$ | $0.627_{\pm 0.0123}$ | $0.630_{\pm 0.0509}$ |
| Frac. Changed | $0.471_{\pm 0.0175}$ | $0.449_{\pm 0.0590}$ | $\mathbf{0.283_{\pm 0.0468}}$ | $0.373_{\pm 0.0123}$ | $0.370_{\pm 0.0509}$ |
| **Top 50 genes** | **Control** | **GEARS** | **scGPT** | **Linear** | **SCCVAE** |
| MSE | $0.00969_{\pm 0.00830}$ | $0.00879_{\pm 0.00770}$ | $0.0215_{\pm 0.00362}$ | $\mathbf{0.00561_{\pm 0.000359}}$ | $0.00651_{\pm 0.00537}$ |
| PearsonR | $0.005_{\pm 0.155}$ | $0.174_{\pm 0.169}$ | $0.464_{\pm 0.107}$ | $0.452_{\pm 0.0426}$ | $\mathbf{0.500_{\pm 0.205}}$ |
| MMD | $0.246_{\pm 0.0512}$ | $0.245_{\pm 0.0531}$ | $0.313_{\pm 0.0246}$ | - | $\mathbf{0.236_{\pm 0.0323}}$ |
| Energy Dist. | $0.0385_{\pm 0.0199}$ | $0.0372_{\pm 0.0203}$ | $0.0588_{\pm 0.0221}$ | - | $\mathbf{0.0338_{\pm 0.0160}}$ |
| Frac. Same | $0.492_{\pm 0.0541}$ | $0.553_{\pm 0.0945}$ | $\mathbf{0.721_{\pm 0.0763}}$ | $0.618_{\pm 0.0163}$ | $0.612_{\pm 0.0820}$ |
| Frac. Changed | $0.468_{\pm 0.0541}$ | $0.447_{\pm 0.0945}$ | $\mathbf{0.279_{\pm 0.0763}}$ | $0.382_{\pm 0.0163}$ | $0.388_{\pm 0.0821}$ |

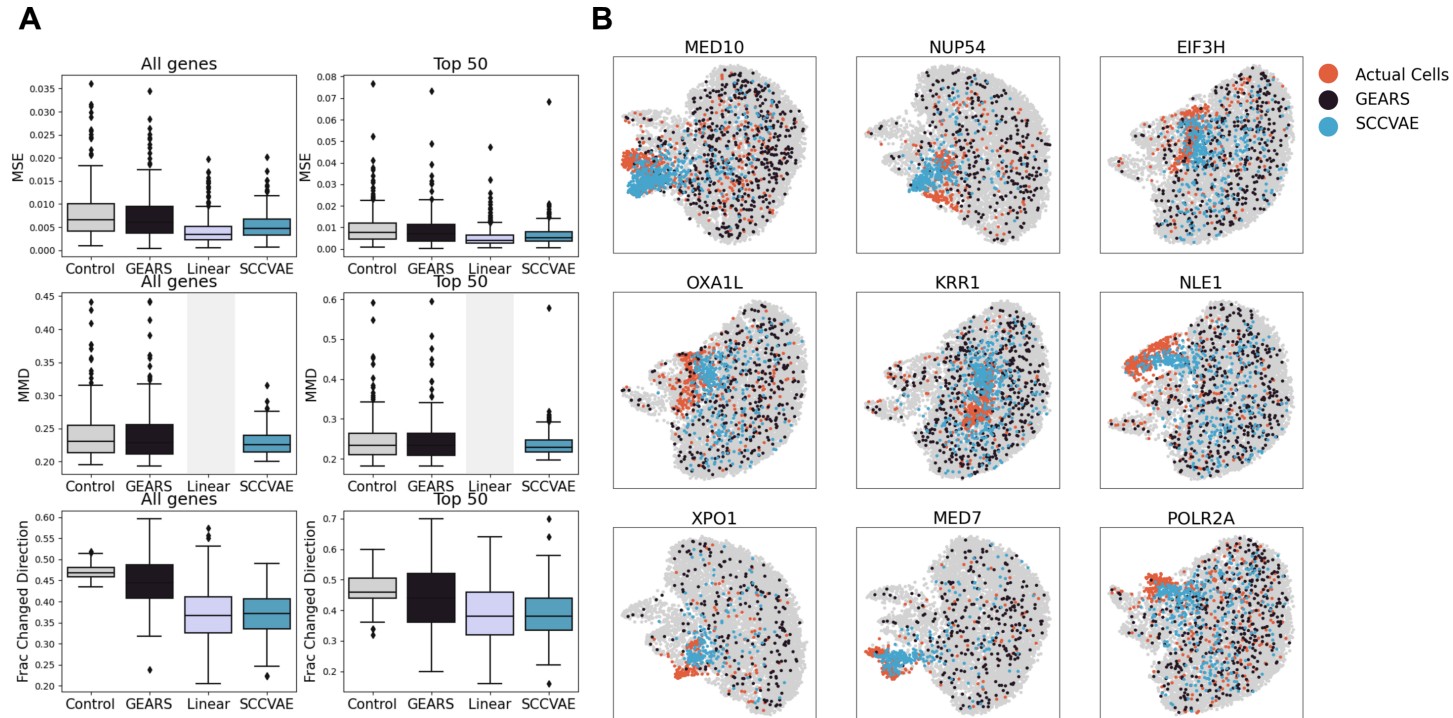

**Fig 2**. **(A) When comparing quantitative performance on the OOD task of GEARS vs SCCVAE to the control distribution, GEARS on average learns the control distribution but SCCVAE is more closely able to approximate the ground truth and is comparable to the linear method across all metrics.** This effect is very pronounced when observing all essential genes, in the case of top 50 genes there are a few outlier perturbations with unusually high error. The linear model is limited to bulk analysis and, therefore, does not include MMD evaluations. (B) SCCVAE and GEARS UMAP visualizations versus ground-truth perturbations on select perturbations in the OOD task. Consistent with quantitative results, GEARS outputs match the control distribution while SCCVAE outputs match the perturbationally distinct ground truth.

models to capture large-scale patterns in the data better than the causal model, but the large distributional error indicates that the model still overfits and cannot generalize to finer distributional details, unlike in the causal model.

**3.2.2 SCCVAE versus linear model.** The performance of the SCCVAE model in comparison to the simple linear model explored in Ahlmann-Eltze et. al is showcased in Table 1 above. ([21]). We use the principal component based representation for the perturbations, same as SCCVAE. It can be seen that both the SCCVAE and the linear model achieve on par performance for average bulk metrics. Recall our discussion in Sect 2.2; notably, the linear model is able to predict the bulk expression well without explicitly specifying the penetrance. This may be because unseen perturbations in this dataset have similar penetrance to those in the training set–specifically, their guides have similar efficiencies. As such, one possible alternative for SCCVAE to perform shift selection in this library can be using predicted bulk expression from the linear model.

On the other hand, the SCCVAE is able to handle single-cell data on a distributional level, while the linear model is limited to bulk analysis. Extending the linear model to capture a distribution over gene expressions is challenging, as it requires specifying higher-order moments of the distribution (beyond just the mean) and modifying the loss function to account for these moments.

## 3.3 Ablation studies

We evaluate the SCCVAE architecture using different choices of graph $\mathcal{G}$ (Sect 2.1). The results in Table 1 are generated using a learned DAG with additional sparsity restrictions (i.e., only upper-triangular mask is applied). We compare the performance using a non-hybrid model with a conditional variational autoencoder (Conditional), a model using a pre-specified gene regulatory network inferred using a causal discovery algorithm on negative control cells [40] (Causal-GSP), and a model using a pre-specified randomly generated graph (Random). Note that these different choices reflect varying sparsity constraints: the conditional variant is equivalent to applying no masking on the DAG; the SCCVAE (with a learned graph) masks only the upper-triangular part of the DAG; the causal-GSP and random variants apply additional masking: beyond the upper-triangular part, they also mask entries in the lower-triangular part that correspond to missing edges. In these cases, causal-GSP uses a graph pre-learned from observational data, whereas the random variant uses a randomly generated graph. We evaluate the performance in the out-of-distribution setting. The details of this setup are described in Note B in S1 Text.

The results of this comparison are shown in Table 2 and Fig 3. We observe that SCCVAE with a learned graph or an inferred DAG identified by GSP outperform the conditional model and the random graph models. Because bulk-level metrics experience minimal change, we focus on MMD. These results further demonstrate that incorporating a mechanistic causal model improves the model performance on generalizing to unseen perturbations.

**3.3.1 SCCVAE versus conditional model.** The causal SCCVAE achieves lower error than the equivalent conditional model, where the input to the decoder is $Z_i + c^p \cdot S_i^p$. The difference is most noticeable in the distributional loss metrics (MMD and Energy Distance), as the conditional model has no way of learning interactions between latent variables that aid in out-of-distribution generalization.

**3.3.2 Learned versus pre-specified causal graphs.** Our results indicate that the sparse graph learned by GSP is overly restrictive in the OOD setting. Overall, it does not generalize as well to the out-of-distribution split as SCCVAE with a learned graph, in terms of the distributional-level metrics (MMD and Energy Distance). However, it is notable that Causal-GSP still performs consistently better than the conditional model, especially on top 50 variable genes, indicating that the sparse graph identified by the GSP algorithm still successfully learns certain regulatory relationships.

**3.3.3 SCCVAE versus random DAGs.** The error metrics (MSE, MMD, Energy Distance) for the random graph model are significantly worse than the other models. However, the average direction of change metrics are improved, likely

**Table 2**. **Ablation studies.** SCCVAE with a learned graph (SCCVAE) consistently achieves superior MSE and MMD values Both the conditional model and the pre-specified graph-based model (Causal-GSP) still outperform the GEARS baselines from Table 1, and the pre-specified graph-based model outperforms the conditional model, but both are more restrictive than SCCVAE with a learned graph. The models with random causal graphs achieve high error, as is expected, but outperform all other models on the fraction same/changed metrics. This is likely due to the shifts being tuned to extreme values during the training/shift selection process to account for the poor distributional match from the random graph.

| All genes | SCCVAE | Conditional | Causal-GSP | Random |
|---|---|---|---|---|
| MSE | **$0.00542_{\pm 0.00295}$** | $0.00580_{\pm 0.00399}$ | $0.00569_{\pm 0.00367}$ | $0.00779_{\pm 0.00270}$ |
| PearsonR | $0.498_{\pm 0.164}$ | $0.435_{\pm 0.206}$ | $0.461_{\pm 0.200}$ | **$0.538_{\pm 0.119}$** |
| MMD | **$0.229_{\pm 0.0193}$** | $0.232_{\pm 0.0293}$ | $0.231_{\pm 0.0271}$ | $0.268_{\pm 0.0259}$ |
| Energy Dist. | **$0.0299_{\pm 0.00958}$** | **$0.0299_{\pm 0.0112}$** | $0.0300_{\pm 0.0104}$ | $0.0332_{\pm 0.00703}$ |
| Frac. Same | $0.630_{\pm 0.0509}$ | $0.618_{\pm 0.0655}$ | $0.622_{\pm 0.0630}$ | **$0.644_{\pm 0.0521}$** |
| Frac. Changed | $0.370_{\pm 0.0509}$ | $0.382_{\pm 0.0655}$ | $0.378_{\pm 0.0630}$ | **$0.356_{\pm 0.0521}$** |
| **Top 50 genes** | **SCCVAE** | **Conditional** | **Causal-GSP** | **Random** |
| MSE | **$0.00651_{\pm 0.00537}$** | $0.00728_{\pm 0.00685}$ | $0.00689_{\pm 0.00641}$ | $0.0101_{\pm 0.00542}$ |
| PearsonR | $0.500_{\pm 0.205}$ | $0.413_{\pm 0.258}$ | $0.474_{\pm 0.238}$ | **$0.519_{\pm 0.178}$** |
| MMD | **$0.236_{\pm 0.0323}$** | $0.243_{\pm 0.0435}$ | $0.240_{\pm 0.0413}$ | $0.289_{\pm 0.0413}$ |
| Energy Dist. | $0.0338_{\pm 0.0160}$ | $0.0329_{\pm 0.0182}$ | **$0.0328_{\pm 0.0167}$** | $0.0389_{\pm 0.013}$ |
| Frac. Same | $0.612_{\pm 0.082}$ | $0.610_{\pm 0.0939}$ | $0.616_{\pm 0.0912}$ | **$0.633_{\pm 0.0717}$** |
| Frac. Changed | $0.388_{\pm 0.0821}$ | $0.390_{\pm 0.0938}$ | $0.384_{\pm 0.0912}$ | **$0.367_{\pm 0.0716}$** |

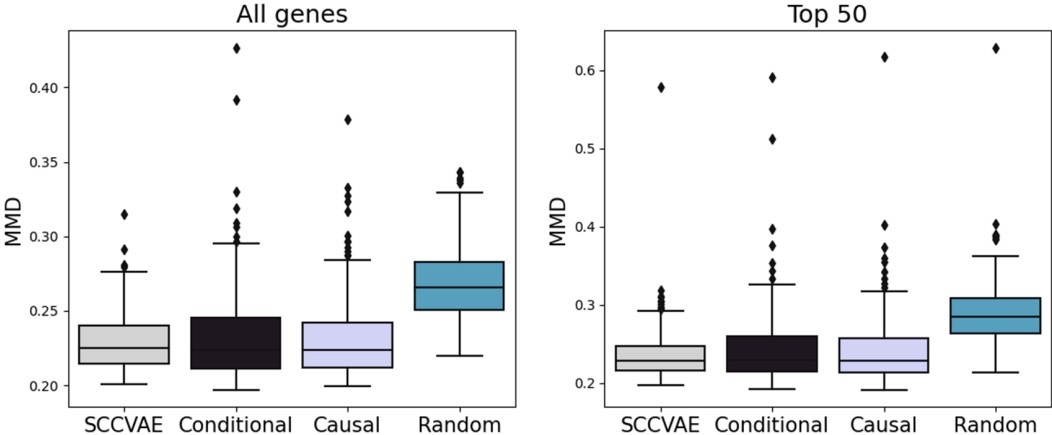

**Fig 3**. **Distribituional loss (MMD) on SCCVAE ablations.** The learned causal graph in the SCCVAE model achieves lower MMD than conditional, sparse causal graph, and random graph equivalents.

because the shift value $c$ (defined in Sect 2.2) is selected to overcompensate for the lack of distributional precision in the predictions. Recall that the shift selection process for $c$ is a required step for out-of-distribution prediction in our mechanistic model, because the penetrance for novel genetic perturbations is unknown. It is worth noting that the subset of pseudo-bulk metrics that the random graph model is able to perform well on are metrics that do not require the predicted numerical values, but only the relative directional change, to match the ground truth closely. The random-graph model imposes a stronger sparsity constraint on the latent causal graph compared to the learned-graph model, which may make these metrics easier to predict. As shown in Table 1, a simple linear model performs quite well on these metrics. However, when the mechanisms in the model are mis-specified (e.g., with a random graph), it is unable to correct the prediction for MSE and distributional metrics. This provides further evidence that the ability of model to generalize to unseen perturbations depends on whether the regulatory information can be successfully learned.

### 3.4 Interpretability of SCCVAE

In this section, we examine the interpretability of SCCVAE, focusing primarily on its learned latent representations. Additional results analyzing the learned structural causal model are provided in Note C.6 in S1 Text. To examine the latent representations learned by SCCVAE, we first investigate if they contain information on whether the perturbations induce distribution shifts from the negative controls. Fig 4 compares the MMD between $X^p$ and $X^{\varnothing}$ in the gene expression space with the $L_2$ distance between $U^p$ and $U^{\varnothing}$ for each $p$ in the latent space in all five OOD splits. For each split, these two distances are highly correlated, which is consistent with the knowledge that SCCVAE learns the perturbation changes through $U^p$, since $U^p = (I_n - A)^{-1}(Z + c^p \cdot S^p)$ and $U^{\varnothing} = (I_n - A)^{-1}Z$ (Sect 2.1) A further breakdown of the scatter plots into training and testing perturbations is found in Note C.3 in S1 Text.

By modulating $c^p$, we can simulate varying penetrance of the perturbation, pushing the post perturbational distribution closer or further away from control. Fig 5 demonstrates this in action on the gene MED7, as $c^p$ is set to various values in $[-1, 3]$. As $c$ is increased from $-1$ to 2, the error decreases until reaching a minimum at $c \approx 2.0$ (Fig 5A). Likewise, the output distribution matches the control distribution when $c^p \approx 0$, most closely matches the ground truth in the dataset at $c^p \approx 2.0$, but can continue to be extended beyond that value as values of $c^p > 2.0$ push the prediction further away from the control distribution (Fig 5B). Further analysis of the effect of shift selection on other perturbation predictions is given in Note C.4 in S1 Text.

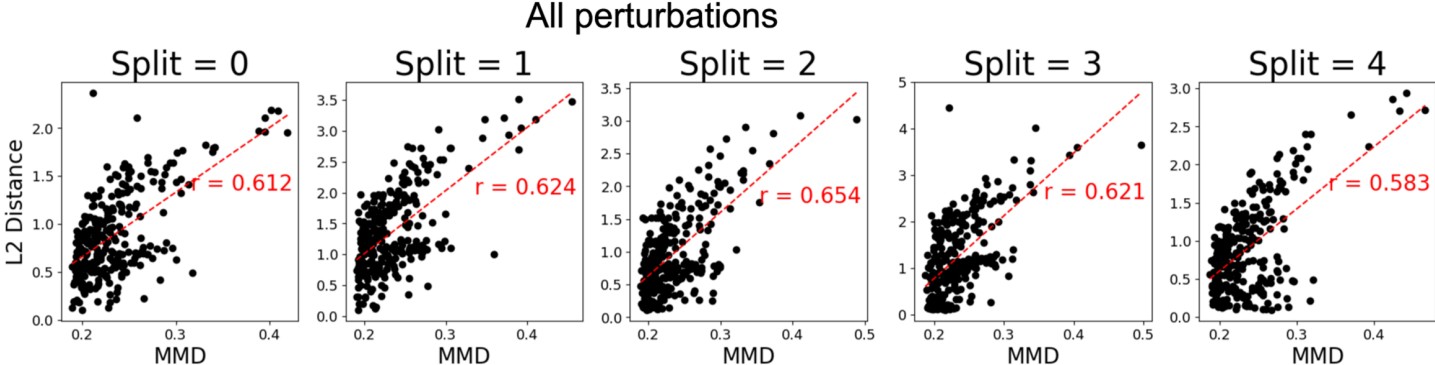

**Fig 4**. **Distance from the control distribution in the observational space vs the latent space ($U^p$), for all perturbations in each OOD split.** The Euclidean distance in the latent space is strongly correlated with the MMD in the expression space.

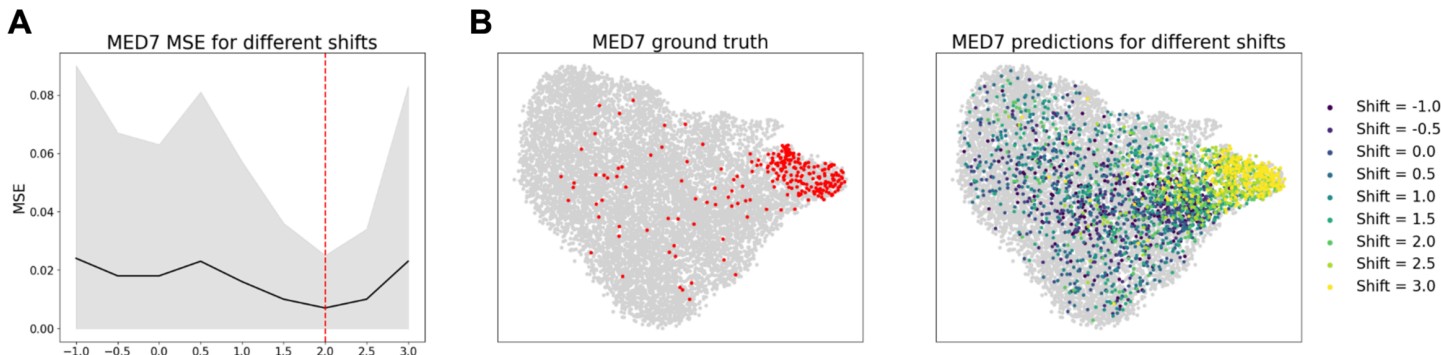

**Fig 5**. **(A) For OOD test perturbations, the shift is selected to minimize MSE of pseudo-bulk predictions.** (B) Shift values closer to zero result in output predictions closer to control, and larger magnitude shift values result in more distinct perturbational distributions. The shift value that most closely matches the ground truth is $c \approx 2.0$.

Fig 6A–6B visualize $U^p$ embeddings for both training and testing distributions from one of the out-of-distribution splits. These embeddings were generated by encoding the perturbations ($p$ or $q$) along with a sample of control cells ($X$), and no reparameterization was used to simulate the effect of taking the average after infinite samples. It can be seen that perturbations of genes with similar functions map to similar variables in the latent space, and identifiable perturbational modules (such as mediator complex genes, ribosomal proteins, cell cycle regulators, proteasome subunits, and genes involved in DNA replication and repair) are clustered together in the latent space, regardless of if the genes in the module were included in the original training set or not. As such, it is possible to infer the function of an unknown gene using SCCVAE by examining the functions of its neighbors when encoded to $U^p$. More visualizations of perturbation modules in $U^p$ can be found in Note C.5 in S1 Text.

## 4 Discussion

In this work, we demonstrated the ability of a hybrid variational causal model (SCCVAE) to effectively model gene expression outcomes following single-gene perturbations, particularly in scenarios when extrapolating to unseen perturbations.

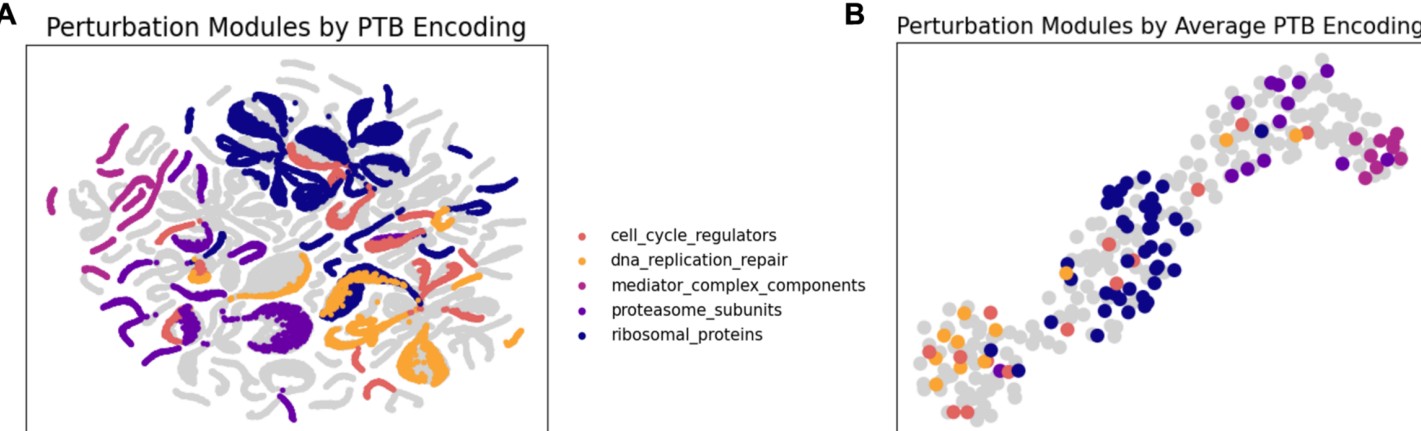

**Fig 6. (A) UMAP Visualizations of latent causal variables with various functional perturbation modules.** Genes belonging to the same perturbation module are close together within the latent space. (B) When visualizing just the average $U^p$ of each perturbation, the perturbation modules form distinct clusters in the latent space.

By leveraging a learned mechanistic model within a flexible variational neural network, SCCVAE is able to both interpolate observed perturbations and generalize to unseen perturbations. These findings underscore the importance of modeling gene regulatory relationships in the predictive model for accurate predictions of genetic interventions. Additionally, the ability of SCCVAE to learn shifts through a structural causal model provides a robust foundation for simulating gene knockdown experiments with varying penetrance.

Our findings open various directions for future exploration. First, we used SCCVAE for single-gene knockdown perturbations. Extending our model to different types of perturbations, or different combinations of perturbations, is an interesting future direction. Second, the current version of SCCVAE implicitly uses a Gaussian likelihood in the output space (i.e., $P(X|Z)$ is modeled as a Gaussian distribution) to approximate the normalized gene expressions. However, raw gene expressions can be modeled using a zero-inflated negative binomial (ZINB) distribution, the parameters of which reveal additional gene-specific information about the perturbed distribution. Extending and further analyzing SCCVAE on ZINB based likelihood is of interest. Lastly, we demonstrated how to incorporate a mechanistic model into a variational autoencoder framework. It would be interesting to build upon this work to test incorporation with alternative generative models.

## Supporting information

**S1 Text. Appendix.**
(PDF)

## Acknowledgments

We would like to acknowledge E. Forte for manuscript proofreading. The authors would like to thank the Eric and Wendy Schmidt Center at the Broad Institute of MIT and Harvard for providing the necessary resources and support for this research.

## Author contributions

**Conceptualization:** Jiaqi Zhang, Caroline Uhler.

**Formal analysis:** Emily Liu, Jiaqi Zhang.

**Funding acquisition:** Caroline Uhler.

**Methodology:** Emily Liu, Jiaqi Zhang, Caroline Uhler.

**Software:** Emily Liu, Jiaqi Zhang.

**Validation:** Emily Liu, Jiaqi Zhang.

**Visualization:** Emily Liu, Jiaqi Zhang.

**Writing – original draft:** Emily Liu, Jiaqi Zhang.

**Writing – review & editing:** Caroline Uhler.

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
