## [Decision Letter · Decision Letter 0]

10 Jul 2025

PCOMPBIOL-D-25-01122

Learning Genetic Perturbation Effects with Variational Causal Inference

PLOS Computational Biology

Dear Dr. Uhler,

Thank you for submitting your manuscript to PLOS Computational Biology. After careful consideration, we feel that it has merit but does not fully meet PLOS Computational Biology's publication criteria as it currently stands. Therefore, we invite you to submit a revised version of the manuscript that addresses the points raised during the review process.

Please submit your revised manuscript within 60 days Sep 09 2025 11:59PM. If you will need more time than this to complete your revisions, please reply to this message or contact the journal office at ploscompbiol@plos.org. Please include the following items when submitting your revised manuscript:

We look forward to receiving your revised manuscript.

Kind regards,

Philipp Martin Altrock, Ph.D.

Academic Editor

PLOS Computational Biology

Marc Birtwistle

Section Editor

PLOS Computational Biology

**Journal Requirements:**

2) Your manuscript is missing the following sections: Discussion.  Please ensure all required sections are present and in the correct order. Make sure section heading levels are clearly indicated in the manuscript text, and limit sub-sections to 3 heading levels. An outline of the required sections can be consulted in our submission guidelines here:

4) Please amend your detailed Financial Disclosure statement. This is published with the article. It must therefore be completed in full sentences and contain the exact wording you wish to be published.

2) If any authors received a salary from any of your funders, please state which authors and which funders..

5)  Please ensure that the funders and grant numbers match between the Financial Disclosure field and the Funding Information tab in your submission form. Note that the funders must be provided in the same order in both places as well.

**Reviewers' comments:**

Reviewer's Responses to Questions

**Comments to the Authors:**

Reviewer #1: The review report is uploaded as an attachment.

Reviewer #2: In this manuscript, the authors present SCCVAE, a deep generative causal inference framework designed to uncover the regulatory network underlying single-cell genetic perturbations. SCCVAE models single-cell gene expression using a structural causal model, where the intrinsic gene modules U_i are assumed to regulate each other through a linear relationship. In the experiments, the authors demonstrated that SCCVAE outperforms baselines like GEARS and scGPT in prediction of single-cell level perturbation effects. Overall the paper is well written, and the proposed method is very interesting. However, there are some questions for the authors to address for publication.

1. (testing on datasets with fewer target genes) In the manuscript, the authors primarily demonstrate the effectiveness of SCCVAE using cell line data with a large number of target genes from Replogle et al. (2022). I wonder how SCCVAE performs when the number of target genes is smaller in the training data, for example, fewer than one hundred?

2. (evaluation) For the comparison, I suggest using the Pearson correlation between predicted and true gene expression changes, instead of between expressions. This is because the correlation between the control cell and perturbed cell gene expression is already very close to 1.

3. (model training) SCCVAE incorporates an additional MMD term during training to align distributions. It would be helpful to provide more details on the implementation of this term (such as how many samples are used for its estimation during training and the choice of kernels) and to demonstrate its contribution to the prediction performance through a simple ablation study.

4. (model interpretability) The authors present the interpretability of the learned SCM in the supplementary notes. However, this is a key strength of SCCVAE that sets it apart from many other perturbation prediction models, and it should be highlighted in the main text. Additionally, it would be interesting to explore differences between the learned graph and a pre-inferred graph using methods like Causal-GSP, beyond just comparing prediction accuracy.

**Have the authors made all data and (if applicable) computational code underlying the findings in their manuscript fully available?**

Reviewer #1: Yes

Reviewer #2: Yes

PLOS authors have the option to publish the peer review history of their article (what does this mean?). If published, this will include your full peer review and any attached files.

Reviewer #1: No

Reviewer #2: No

**Figure resubmission:**
---

## [Decision Letter · Decision Letter 1]

29 Sep 2025

PCOMPBIOL-D-25-01122R1

Learning Genetic Perturbation Effects with Variational Causal Inference

PLOS Computational Biology

Dear Dr. Uhler,

Thank you for submitting your manuscript to PLOS Computational Biology. After careful consideration, we feel that it has merit but does not fully meet PLOS Computational Biology's publication criteria as it currently stands. Therefore, we invite you to submit a revised version of the manuscript that addresses the points raised during the review process.

Please submit your revised manuscript within 60 days Nov 29 2025 11:59PM. If you will need more time than this to complete your revisions, please reply to this message or contact the journal office at ploscompbiol@plos.org. Please include the following items when submitting your revised manuscript:

We look forward to receiving your revised manuscript.

Kind regards,

Philipp Martin Altrock, Ph.D.

Academic Editor

PLOS Computational Biology

Marc Birtwistle

Section Editor

PLOS Computational Biology

**Journal Requirements:**

1) We note that your Manuscript files and Supplementary Figures files are duplicated on your submission. The Supplementary figures are uploaded separately and are also part of your (SSCCVAE_revised_supplement.pdf). Please remove any unnecessary or old files from your revision, and make sure that only those relevant to the current version of the manuscript are included.

2) We have noticed that you have uploaded Supporting Information files, but you have not included a list of legends. Please add a full list of legends for your Supporting Information files after the references list.

3) Please cite and label the supplementary tables and figures as “S1 Table” and “S2 Table,” "S1 Figure", S2 Figure" and so forth.

**Reviewers' comments:**

Reviewer's Responses to Questions

Reviewer #1: The authors' thorough rebuttal and revision have addressed my main concerns. I have only one very minor suggestion. There is a recently published work also building causality into modern AI-type models (diffusion models) with the hope to get better generalization in comp-bio problems: https://www.nature.com/articles/s41467-025-63728-0. It is recommended to also briefly discuss this work, in terms of the differences in focus, assumptions, and settings.

Reviewer #2: In their response, the authors have addressed most of my concerns and substantially clarified the paper. However, based on the revised results, one issue remains confusing: the use of random graphs often achieves better PearsonR, which is counterintuitive. This suggests that a misspecified causal structure among gene modules might somehow improve perturbation effect prediction.

I also noticed that, in the ablation studies, the DAG was learned without sparsity constraints, whereas in the main method the authors reported that using an upper-triangular mask worked well. The authors should explain why the DAG structures are inconsistent and clarify how different choices of DAG parameterizations contribute to improving the model.

Finally, in Section 3.3.3, the statement “The error metrics (MSE, Pearson, MMD, Energy Distance) for the random graph model are significantly worse than the other models” should be revised considering the updated results.

**Have the authors made all data and (if applicable) computational code underlying the findings in their manuscript fully available?**

Reviewer #1: Yes

Reviewer #2: Yes

PLOS authors have the option to publish the peer review history of their article (what does this mean?). If published, this will include your full peer review and any attached files.

Reviewer #1: No

Reviewer #2: No

**Figure resubmission:**
---

## [Decision Letter · Decision Letter 2]

6 Jan 2026

Dear Dr. Uhler,

We are pleased to inform you that your manuscript 'Learning Genetic Perturbation Effects with Variational Causal Inference' has been provisionally accepted for publication in PLOS Computational Biology.

Best regards,

Philipp Martin Altrock, Ph.D.

Academic Editor

PLOS Computational Biology

Marc Birtwistle

Section Editor

PLOS Computational Biology

Reviewer's Responses to Questions

**Comments to the Authors:**

Reviewer #2: The authors have addressed my concerns.

**Have the authors made all data and (if applicable) computational code underlying the findings in their manuscript fully available?**

Reviewer #2: None

PLOS authors have the option to publish the peer review history of their article (what does this mean?). If published, this will include your full peer review and any attached files.

Reviewer #2: No

---

## [Editor Report · Acceptance letter]

PCOMPBIOL-D-25-01122R2

Learning Genetic Perturbation Effects with Variational Causal Inference

Dear Dr Uhler,

I am pleased to inform you that your manuscript has been formally accepted for publication in PLOS Computational Biology. Your manuscript is now with our production department and you will be notified of the publication date in due course.

With kind regards,

Anita Estes
